# Bi-Directional Block Self-Attention for Fast and Memory-Efficient Sequence Modeling

**Tao Shen†, Tianyi Zhou‡, Guodong Long†, Jing Jiang†& Chengqi Zhang†**
†Centre for Artificial Intelligence, School of Software, University of Technology Sydney
‡Paul G. Allen School of Computer Science & Engineering, University of Washington
`tao.shen@student.uts.edu.au,tianyizh@uw.edu`
`{guodong.long,jing.jiang,chengqi.zhang}@uts.edu.au`

## Abstract

Recurrent neural networks (RNN), convolutional neural networks (CNN) and self-attention networks (SAN) are commonly used to produce context-aware representations. RNN can capture long-range dependency but is hard to parallelize and not time-efficient. CNN focuses on local dependency but does not perform well on some tasks. SAN can model both such dependencies via highly parallelizable computation, but memory requirement grows rapidly in line with sequence length. In this paper, we propose a model, called "bi-directional block self-attention network (Bi-BloSAN)", for RNN/CNN-free sequence encoding. It requires as little memory as RNN but with all the merits of SAN. Bi-BloSAN splits the entire sequence into blocks, and applies an intra-block SAN to each block for modeling local context, then applies an inter-block SAN to the outputs for all blocks to capture long-range dependency. Thus, each SAN only needs to process a short sequence, and only a small amount of memory is required. Additionally, we use feature-level attention to handle the variation of contexts around the same word, and use forward/backward masks to encode temporal order information. On nine benchmark datasets for different NLP tasks, Bi-BloSAN achieves or improves upon state-of-the-art accuracy, and shows better efficiency-memory trade-off than existing RNN/CNN/SAN.

## 1 Introduction

Context dependency provides critical information for most natural language processing (NLP) tasks. In deep neural networks (DNN), context dependency is usually modeled by a context fusion module, whose goal is to learn a context-aware representation for each token from the input sequence. Recurrent neural networks (RNN), convolutional neural networks (CNN) and self-attention networks (SAN) are commonly used as context fusion modules. However, each has its own merits and defects, so which network to use is an open problem and mainly depends on the specific task.

RNN is broadly used given its capability in capturing long-range dependency through recurrent computation. It has been applied to various NLP tasks, e.g., question answering (Wang et al., 2017), neural machine translation (Bahdanau et al., 2015), sentiment analysis (Qian et al., 2017), natural language inference (Liu et al., 2016), etc. However, training the basic RNN encounters the gradient dispersion problem, and is difficult to parallelize. Long short-term memory (LSTM) (Hochreiter & Schmidhuber, 1997) effectively avoids the vanishing gradient. Gated recurrent unit (GRU) (Chung et al., 2014) and simple recurrent unit (SRU) (Lei & Zhang, 2017) improve the efficiency by reducing parameters and removing partial temporal-dependency, respectively. However, they still suffer from expensive time cost, especially when applied to long sequences.

CNN becomes popular recently on some NLP tasks because of its the highly parallelizable convolution computation (Dong et al., 2017). Unlike RNN, CNN can simultaneously apply convolutions defined by different kernels to multiple chunks of a sequence (Kim, 2014). It is mainly used for sentence-encoding tasks (Lei et al., 2015; Kalchbrenner et al., 2014). Recently, hierarchical CNNs, e.g. ByteNet (Kalchbrenner et al., 2016), and ConvS2S (Gehring et al., 2017), are proposed to capture relatively long-range dependencies by using stacking CNNs to increase the number of input

elements represented in a state. Nonetheless, as mentioned by Vaswani et al. (2017), the number of CNNs required to relate signals from two arbitrary input grows in the distance between positions, linearly for ConvS2S and logarithmically for ByteNet. This makes it difficult to learn dependencies between distant positions.

Recently, self-attention networks (SAN) have been successfully applied to several NLP tasks. It produces context-aware representation by applying attention to each pair of tokens from the input sequence. Compared to RNN/CNN, SAN is flexible in modeling both long-range and local dependencies. The major computation in SAN is the highly parallelizable matrix multiplication without any temporal iteration, which can be easily accelerated by existing tools. Unlike most works that attach SAN to RNN/CNN as an additional module, two recent works show that SAN independent of any RNN/CNN module can achieve state-of-the-art performance on several NLP tasks. The first, multi-head attention (Vaswani et al., 2017), is a major component of a seq2seq model "Transformer" that outperforms previous methods in neural machine translation. It projects the input sequence into multiple subspaces, applies a SAN to the representation in each subspace, and concatenates the outputs. The second, directional self-attention network (DiSAN) (Shen et al., 2017), computes alignment scores at feature level, rather than at token level, and applies forward/backward masks to the alignment score matrix to encode temporal order information. DiSAN achieves the best or state-of-the-art test accuracy on several NLP tasks by using less computational time and fewer parameters. More related works can be found in Appendix D.

However, one drawback of SAN is its large memory requirement to store the alignment scores of all the token pairs; the number grows quadratically with the sequence length. By contrast, RNN/CNN demand far less memory. The goal of this paper is to develop a novel SAN for RNN/CNN-free sequence encoding, which requires as little memory as RNN but inherits all the advantages of SAN, i.e., highly parallelizable computation, the capability/flexibility in modeling both long-range/local dependencies, and state-of-the-art performance on multiple NLP tasks.

We propose an attention mechanism, called "bi-directional block self-attention (Bi-BloSA)", for fast and memory-efficient context fusion. The basic idea is to split a sequence into several length-equal blocks (with padding if necessary), and apply an intra-block SAN to each block independently. The outputs for all the blocks are then processed by an inter-block SAN. The intra-block SAN captures the local dependency within each block, while the inter-block SAN captures the long-range/global dependency. Hence, every SAN only needs to process a short sequence. Compared to a single SAN applied to the whole sequence, such two-layer stacked SAN saves a significant amount of memory. A feature fusion gate combines the outputs of intra-block and inter-block SAN with the original input, to produce the final context-aware representations of all the tokens. Similar to directional self-attention (DiSA) (Shen et al., 2017), Bi-BloSA uses forward/backward masks to encode the temporal order information, and feature-level attention to handle the variation of contexts around the same word. Further, a RNN/CNN-free sequence encoding model we build based on Bi-BloSA, called "bi-directional block self-attention network (Bi-BloSAN)", uses an attention mechanism to compress the output of Bi-BloSA into a vector representation.

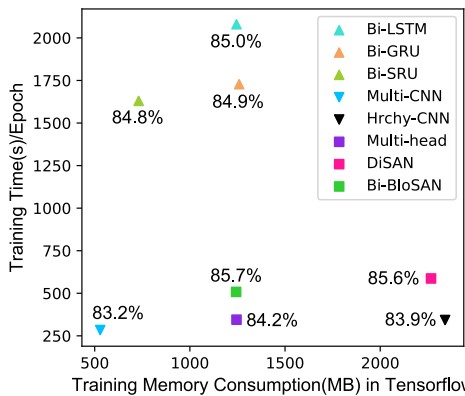

Figure 1: A comparison of Bi-BloSAN and other RNN/CNN/SAN in terms of **training time**, **training memory consumption** and **test accuracy** on SNLI (Bowman et al., 2015). The details of all the models are provided in Section 4.

In experiments[1], we implement Bi-BloSAN and popular sequence encoding models on several NLP tasks, e.g., language inference, sentiment analysis, semantic relatedness, reading comprehension, question-type classification, etc. The baseline models include Bi-LSTM, Bi-GRU, Bi-SRU, CNNs, multi-head attention and DiSAN. A thorough comparison on nine benchmark datasets demonstrates the advantages of Bi-BloSAN in terms of training speed, inference accuracy and memory consumption. Figure 1 shows that Bi-BloSAN obtains the best accuracy by costing similar training time

---

[1] Source code and scripts for experiments are at https://github.com/taoshen58/BiBloSA

to DiSAN, and as little memory as Bi-LSTM, Bi-GRU and multi-head attention. This shows that Bi-BloSAN achieves a better efficiency-memory trade-off than existing RNN/CNN/SAN models.

Our notations follow these conventions: 1) lowercase denotes a vector; 2) bold lowercase denotes a sequence of vectors (stored as a matrix); and 3) uppercase denotes a matrix or a tensor.

## 2 BACKGROUND

### 2.1 WORD EMBEDDING

Word embedding is the basic processing unit in most DNN for sequence modeling. It transfers each discrete token into a representation vector of real values. Given a sequence of tokens (e.g., words or characters) $\boldsymbol{w} = [w_1, w_2, \ldots, w_n] \in \mathbb{R}^{N \times n}$, where $w_i$ is a one-hot vector, $N$ is the vocabulary size and $n$ is the sequence length. A pre-trained token embedding (e.g. word2vec (Mikolov et al., 2013b)) is applied to $\boldsymbol{w}$, which outputs a sequence of low dimensional vectors $\boldsymbol{x} = [x_1, x_2, \ldots, x_n] \in \mathbb{R}^{d_e \times n}$. This process can be formally written as $\boldsymbol{x} = W^{(e)}\boldsymbol{w}$, where $W^{(e)} \in \mathbb{R}^{d_e \times N}$ is the embedding weight matrix that can be fine-tuned during the training phase.

### 2.2 VANILLA ATTENTION AND MULTI-DIMENSIONAL ATTENTION

**Vanilla Attention**: Given an input sequence $\boldsymbol{x} = [x_1, x_2, \ldots, x_n]$ composed of token embeddings and a vector representation of a query $q \in \mathbb{R}^{d_q}$, vanilla attention (Bahdanau et al., 2015) computes the alignment score between $q$ and each token $x_i$ (reflecting the attention of $q$ to $x_i$) using a compatibility function $f(x_i, q)$. A softmax function then transforms the alignment scores $a \in \mathbb{R}^n$ to a probability distribution $p(z|\boldsymbol{x}, q)$, where $z$ is an indicator of which token is important to $q$. A large $p(z = i|\boldsymbol{x}, q)$ means that $x_i$ contributes important information to $q$. This process can be written as

$$a = [f(x_i, q)]_{i=1}^{n}, \tag{1}$$

$$p(z|\boldsymbol{x}, q) = \mathrm{softmax}(a). \tag{2}$$

The output $s$ is the expectation of sampling a token according to its importance, i.e.,

$$s = \sum_{i=1}^{n} p(z = i|\boldsymbol{x}, q)x_i = \mathbb{E}_{i \sim p(z|\boldsymbol{x}, q)}(x_i). \tag{3}$$

Multiplicative attention (or dot-product attention) (Vaswani et al., 2017; Sukhbaatar et al., 2015; Rush et al., 2015) and additive attention (or multi-layer perceptron attention) (Bahdanau et al., 2015; Shang et al., 2015) are two commonly used attention mechanisms. They differ in the choice of compatibility function $f(x_i, q)$. Multiplicative attention uses the cosine similarity for $f(x_i, q)$, i.e.,

$$f(x_i, q) = \left\langle W^{(1)}x_i, W^{(2)}q \right\rangle, \tag{4}$$

where $W^{(1)} \in \mathbb{R}^{d_h \times d_e}$, $W^{(2)} \in \mathbb{R}^{d_h \times d_q}$ are the learnable parameters. Additive attention is defined as

$$f(x_i, q) = w^T \sigma(W^{(1)}x_i + W^{(2)}q + b^{(1)}) + b, \tag{5}$$

where $w \in \mathbb{R}^{d_h}$, $b^{(1)}$ and $b$ are the biases, and $\sigma(\cdot)$ is an activation function. Additive attention usually achieves better empirical performance than multiplicative attention, but is expensive in time cost and memory consumption.

**Multi-dimensional Attention**: Unlike vanilla attention, in multi-dimensional (multi-dim) attention (Shen et al., 2017), the alignment score is computed for each feature, i.e., the score of a token pair is a vector rather than a scalar, so the score might be large for some features but small for others. Therefore, it is more expressive than vanilla attention, especially for the words whose meaning varies in different contexts.

Multi-dim attention has $d_e$ indicators $z_1, \ldots, z_{d_e}$ for $d_e$ features. Each indicator has a probability distribution that is generated by applying $\mathrm{softmax}$ to the $n$ alignment scores of the corresponding feature. Hence, for each feature $k$ in each token $i$, we have $P_{ki} \triangleq p(z_k = i|\boldsymbol{x}, q)$ where $P \in \mathbb{R}^{d_e \times n}$.

A large $P_{ki}$ means that the feature $k$ in token $i$ is important to $q$. The output of multi-dim attention is written as

$$s = \left[\sum_{i=1}^{n} P_{ki} \boldsymbol{x}_{ki}\right]_{k=1}^{d_e} = \left[\mathbb{E}_{i \sim p(z_k|\boldsymbol{x},q)}(\boldsymbol{x}_{ki})\right]_{k=1}^{d_e}. \tag{6}$$

For simplicity, we ignore the subscript $k$ where no confusion is caused. Then, Eq.(6) can be rewritten as an element-wise product, i.e., $s = \sum_{i=1}^{n} P_{\cdot i} \odot x_i$. Here, $P_{\cdot i}$ is computed by the additive attention in Eq.(5) where $w^T$ is replaced with a weight matrix $W \in \mathbb{R}^{d_h \times d_e}$, which leads to a score vector for each token pair.

## 2.3 TWO TYPES OF SELF-ATTENTION

**token2token self-attention** (Hu et al., 2017; Vaswani et al., 2017; Shen et al., 2017) produces context-aware representations by exploring the dependency between two tokens $x_i$ and $x_j$ from the same sequence $\boldsymbol{x}$. In particular, $q$ in Eq.(5) is replaced with $x_j$, i.e.,

$$f(x_i, x_j) = W^T \sigma(W^{(1)} x_i + W^{(2)} x_j + b^{(1)}) + b. \tag{7}$$

Similar to the $P$ in multi-dim attention, each input token $x_j$ is associated with a probability matrix $P^j$ such that $P_{ki}^j \triangleq p(z_k = i | \boldsymbol{x}, x_j)$. The output representation for $x_j$ is $s_j = \sum_{i=1}^{n} P_{\cdot i}^j \odot x_i$ and the final output of token2token self-attention is $\boldsymbol{s} = [s_1, s_2, \ldots, s_n]$.

**source2token self-attention** (Lin et al., 2017; Shen et al., 2017; Liu et al., 2016) explores the importance of each token to the entire sentence given a specific task. In particular, $q$ is removed from Eq.(5), and the following equation is used as the compatibility function.

$$f(x_i) = W^T \sigma(W^{(1)} x_i + b^{(1)}) + b. \tag{8}$$

The probability matrix $P$ is defined as $P_{ki} \triangleq p(z_k = i | \boldsymbol{x})$. The final output of source2token self-attention has the same form as multi-dim attention, i.e., $s = \sum_{i=1}^{n} P_{\cdot i} \odot x_i$

## 2.4 MASKED SELF-ATTENTION

Temporal order information is difficult to encode in token2token self-attention introduced above because the alignment score between two tokens is symmetric. Masked self-attention (Shen et al., 2017) applies a mask $M \in \mathbb{R}^{n \times n}$ to the alignment score matrix (or tensor due to feature-level score) computed by Eq.(7), so it allows one-way attention from one token to another. Specifically, the bias $b$ in Eq.(7) is replaced with a constant vector $M_{ij}\mathbf{1}$, where the $\mathbf{1}$ is an all-one vector. In addition, $W$ is fixed to a scalar $c$ and $\tanh(\cdot/c)$ is used as the activation function $\sigma(\cdot)$, i.e.,

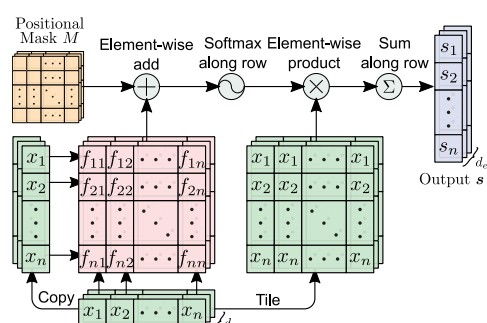

Figure 2: Masked self-attention mechanism. $f_{ij}$ denotes $f(x_i, x_j)$ in Eq.(9).

$$f(x_i, x_j) = c \cdot \tanh\left([W^{(1)} x_i + W^{(2)} x_j + b^{(1)}]/c\right) + M_{ij}\mathbf{1}, \tag{9}$$

where $W^{(1)} \in \mathbb{R}^{d_e \times d_e}, W^{(2)} \in \mathbb{R}^{d_e \times d_q}$. The procedures to calculate the attention output from $f(x_i, x_j)$ are identical to those in token2token self-attention. We use $\boldsymbol{s} = g^m(\boldsymbol{x}, M)$ to denote the complete process of masked self-attention with $\boldsymbol{s} = [s_1, s_2, \ldots, s_n]$ as the output sequence. An illustration of masked self-attention is given in Figure 2.

In order to model bi-directional order information, forward mask $M^{fw}$ and backward mask $M^{bw}$ are respectively substituted into Eq.(9), which results in forward and backward self-attentions. These two masks are defined as

$$M_{ij}^{fw} = \begin{cases} 0, & i < j \\ -\infty, & \text{otherwise} \end{cases} \qquad M_{ij}^{bw} = \begin{cases} 0, & i > j \\ -\infty, & \text{otherwise} \end{cases} \tag{10}$$

The outputs of forward and backward self-attentions are denoted by $\boldsymbol{s^{fw}} = g^m(\boldsymbol{x}, M^{fw})$ and $\boldsymbol{s^{bw}} = g^m(\boldsymbol{x}, M^{bw})$, respectively.

## 3 PROPOSED MODEL

In this section, we first introduce the "masked block self-attention (mBloSA)" (Section 3.1) as a fundamental self-attention module. Then, we present the "bi-directional block self-attention network (Bi-BloSAN)" (Section 3.2) for sequence encoding, which uses the "bi-directional block self-attention (Bi-BloSA)" (mBloSA with forward and backward masks) as its context fusion module.

### 3.1 MASKED BLOCK SELF-ATTENTION

As shown in Figure 3, masked block self-attention (mBloSA) has three parts from its bottom to top, i.e., 1) intra-block self-attention, 2) inter-block self-attention, and 3) the context fusion.

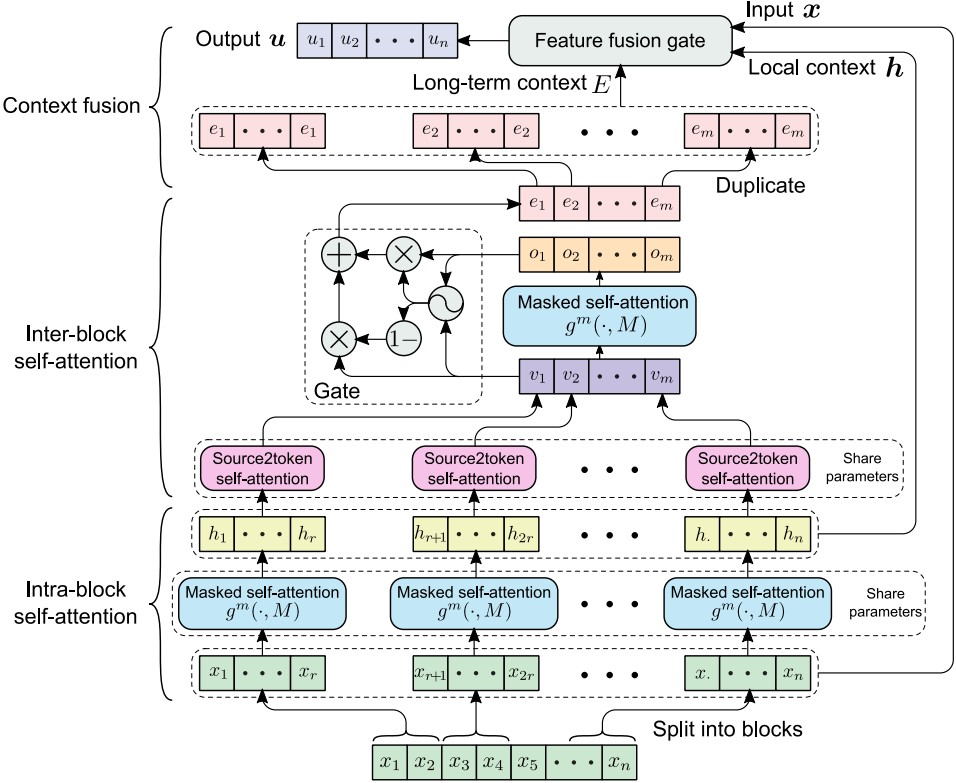

Figure 3: Masked block self-attention (mBloSA) mechanism.

**Intra-block self-attention**: We firstly split the input sequence of token/word embeddings into $m$ blocks of equal length $r$, i.e., $[\boldsymbol{x^l}]_{l=1}^m = [\boldsymbol{x^1}, \boldsymbol{x^2}, \ldots, \boldsymbol{x^m}]$ where $\boldsymbol{x^1} = [x_1, x_2, \ldots, x_r]$, $\boldsymbol{x^2} = [x_{r+1}, x_{r+2}, \ldots, x_{2r}]$ and $\boldsymbol{x^m} = [x_{n-r+1}, x_{n-r+2}, \ldots, x_n]$. Padding can be applied to the last block if necessary. Intra-block self-attention applies the masked self-attentions $g^m(\cdot, M)$ with shared parameters to all the blocks , i.e.,

$$\boldsymbol{h^l} = g^m(\boldsymbol{x^l}, M),\ l = 1, 2, \ldots, m. \tag{11}$$

Its goal is to capture the local context dependency inside each block. Similar to $\boldsymbol{x^l}$, the output representations of the tokens in the $l$-th block are denoted by $\boldsymbol{h^l} = [h_{r(l-1)+1}, h_{r(l-1)+2}, \ldots, h_{r \times l}]$. Note, the block length $r$ is a hyper-parameter and $m = n/r$. In Appendix A, we introduce an approach to selecting the optimal $r$, which results in the maximum memory utility rate in expectation.

**Inter-block self-attention**: To generate a vector representation $v^l$ of each block, a source2token self-attention $g^{s2t}(\cdot)$ is applied to the output $\boldsymbol{h^l}$ of the intra-block self-attention on each block, i.e.,

$$v^l = g^{s2t}(\boldsymbol{h^l}),\ l = 1, 2, \ldots, m. \tag{12}$$

Note we apply the parameter-shared $g^{s2t}(\cdot)$ to $\boldsymbol{h^l}$ for different blocks. This provides us with a sequence $\boldsymbol{v} = [v_1, v_2, \ldots, v_m]$ of local-context representations at block level. Inter-block self-

attention then applies a masked self-attention to $\boldsymbol{v}$ in order to capture the long-range/global dependency among the blocks, i.e.,

$$\boldsymbol{o} = g^m(\boldsymbol{v}, M). \tag{13}$$

To combine the local and global context features at block level, a gate is used to merge the input and the output of the masked self-attention dynamically. This is similar to the gates in LSTM. The output sequence $\boldsymbol{e} = [e_1, \ldots, e_m]$ of the gate is computed by

$$G = \text{sigmoid}\left(W^{(g1)}\boldsymbol{o} + W^{(g2)}\boldsymbol{v} + b^{(g)}\right), \tag{14}$$

$$\boldsymbol{e} = G \odot \boldsymbol{o} + (1 - G) \odot \boldsymbol{v} \tag{15}$$

**Context fusion**: Given the long-range context representations $\boldsymbol{e} = [e_1, \ldots, e_m] \in \mathbb{R}^{d_e \times m}$ at block level, we duplicate $e_l$ for $r$ times to get $\boldsymbol{e^l} = [e_l, e_l, \ldots, e_l]$ (each token in block $l$ has the global context feature representation $e_l$). Let $E \triangleq [\boldsymbol{e^l}]_{l=1}^m \in \mathbb{R}^{d_e \times n}$. Now, we have the input sequence $\boldsymbol{x}$ of word embeddings, the local context features $\boldsymbol{h}$ produced by intra-block self-attention, and the long-range/global context features $E$ produced by inter-block self-attention. A feature fusion gate (Gong & Bowman, 2017) is employed to combine them, and generates the final context-aware representations of all tokens, i.e.,

$$F = \sigma\left(W^{(f1)}[\boldsymbol{x}; \boldsymbol{h}; E] + b^{(f1)}\right), \tag{16}$$

$$G = \text{sigmoid}\left(W^{(f2)}[\boldsymbol{x}; \boldsymbol{h}; E] + b^{(f2)}\right), \tag{17}$$

$$\boldsymbol{u} = G \odot F + (1 - G) \odot \boldsymbol{x}, \tag{18}$$

where $\sigma(\cdot)$ is an activation function, and $\boldsymbol{u} = [u_1, u_2, \ldots, u_n] \in \mathbb{R}^{d_e \times n}$ is the mBloSA output, which consists of the context-aware representations of the $n$ tokens.

### 3.2 BI-DIRECTIONAL BLOCK SELF-ATTENTION NETWORK FOR SEQUENCE ENCODING

We propose a sequence encoding model "Bi-directional block self-attention network (Bi-BloSAN)" with mBloSA as its major components. Its architecture is shown in Figure 4. In Bi-BloSAN, two fully connected layers (with untied parameters) are applied to the input sequence of token embeddings. Their outputs are processed by two mBloSA modules respectively. One uses the forward mask $M^{fw}$ and another uses the backward mask $M^{bw}$. Their outputs $\boldsymbol{u^{fw}}$ and $\boldsymbol{u^{bw}}$ are concatenated as $\boldsymbol{u^{bi}} = [\boldsymbol{u^{fw}}; \boldsymbol{u^{bw}}] \in \mathbb{R}^{2d_e \times n}$. The idea of bi-directional attention follows the same spirit as Bi-LSTM and DiSAN. It

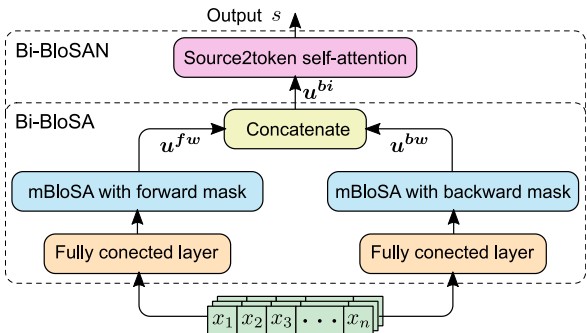

Figure 4: Bi-directional block self-attention network (Bi-BloSAN) for sequence encoding.

encodes temporal order information lacking in existing SAN models. The context fusion module in Bi-BloSAN, with the input $\boldsymbol{x}$ and the output $\boldsymbol{u^{bi}}$, is called "Bi-BloSA". In order to obtain a sequence encoding, a source2token self-attention transforms the sequence $\boldsymbol{u^{bi}}$ of concatenated token representations into a vector representation $s$.

## 4 EXPERIMENTS

We conduct the experiments of Bi-BloSAN and several popular RNN/CNN/SAN-based sequence encoding models on nine benchmark datasets for multiple different NLP tasks. Note that, in some baseline models, a source2token self-attention is on the top of the models to generate an encoding for the entire sequence. All the models used for comparisons are listed as follows.

- **Bi-LSTM**: 600D Bi-directional LSTM (300D forward LSTM + 300D backward LSTM) (Graves et al., 2013).

- **Bi-GRU**: 600D Bi-directional GRU (Chung et al., 2014).
- **Bi-SRU**: 600D Bi-directional SRU (Lei & Zhang, 2017) (with sped-up recurrence but no CUDA level optimization for fair comparison).
- **Multi-CNN**: 600D CNN sentence embedding model (Kim, 2014) (200D for each of 3, 4, 5-gram).
- **Hrchy-CNN**: 3-layer 300D CNN (Gehring et al., 2017) with kernel length 5, to which gated linear units (Dauphin et al., 2016) and residual connection (He et al., 2016) are applied.
- **Multi-head**: 600D Multi-head attention (Vaswani et al., 2017) (8 heads, each has 75 hidden units). The positional encoding method used in Vaswani et al. (2017) is applied to the input sequence to encode temporal order information.
- **DiSAN**: 600D Directional self-attention network (Shen et al., 2017) (300D forward masked self-attention + 300D backward masked self-attention).

All experimental codes are implemented in Python with Tensorflow and run on a single Nvidia GTX 1080Ti graphic card. Both time cost and memory load data are collected under Tensorflow1.3 with CUDA8 and cuDNN6021. In the rest of this section, we conduct the experiments on natural language inference in Section 4.1, reading comprehension in Section 4.2, semantic relatedness in Section 4.3 and sentence classifications in Section 4.4. Finally, we analyze the time cost and memory load of the different models vs. the sequence length in Section 4.5.

## 4.1 NATURAL LANGUAGE INFERENCE

Natural language inference (NLI) aims to reason the semantic relationship between a pair of sentences, i.e., a premise sentence and a hypothesis sentence. This relationship could be *entailment*, *neutral* or *contradiction*. In the experiment, we compare Bi-BloSAN to other baselines on the Stanford Natural Language Inference (Bowman et al., 2015) (SNLI)[2] dataset, which contains standard training/dev/test split of 549,367/9,842/9,824 samples.

Table 1: Experimental results for different methods on SNLI. $|\theta|$: the number of parameters (excluding word embedding part). **Train Accu** and **Test Accu**: the accuracies on training and test sets respectively.

| Model | $|\theta|$ | Train Accu | Test Accu |
|---|---|---|---|
| Unlexicalized features (Bowman et al., 2015) | | 49.4 | 50.4 |
| + Unigram and bigram features (Bowman et al., 2015) | | 99.7 | 78.2 |
| 100D LSTM encoders (Bowman et al., 2015) | 0.2m | 84.8 | 77.6 |
| 300D LSTM encoders (Bowman et al., 2016) | 3.0m | 83.9 | 80.6 |
| 1024D GRU encoders (Vendrov et al., 2016) | 15.0m | 98.8 | 81.4 |
| 300D Tree-based CNN encoders (Mou et al., 2016) | 3.5m | 83.3 | 82.1 |
| 300D SPINN-PI encoders (Bowman et al., 2016) | 3.7m | 89.2 | 83.2 |
| 600D Bi-LSTM encoders (Liu et al., 2016) | 2.0m | 86.4 | 83.3 |
| 300D NTI-SLSTM-LSTM encoders (Munkhdalai & Yu, 2017b) | 4.0m | 82.5 | 83.4 |
| 600D Bi-LSTM encoders+intra-attention (Liu et al., 2016) | 2.8m | 84.5 | 84.2 |
| 300D NSE encoders (Munkhdalai & Yu, 2017a) | 3.0m | 86.2 | 84.6 |
| 600D (300+300) Deep Gated Attn. (Chen et al., 2017) | 11.6m | 90.5 | 85.5 |
| Bi-LSTM (Graves et al., 2013) | 2.9m | 90.4 | 85.0 |
| Bi-GRU (Chung et al., 2014) | 2.5m | 91.9 | 84.9 |
| Bi-SRU (Lei & Zhang, 2017) | 2.0m | 88.4 | 84.8 |
| Multi-CNN (Kim, 2014) | 1.4m | 89.3 | 83.2 |
| Hrchy-CNN (Gehring et al., 2017) | 3.4m | 91.3 | 83.9 |
| Multi-head (Vaswani et al., 2017) | 2.0m | 89.6 | 84.2 |
| DiSAN (Shen et al., 2017) | 2.3m | 91.1 | 85.6 |
| 480D Bi-BloSAN | 2.8m | 91.7 | **85.7** |

Following the method of applying sentence-encoding to NLI task given in Bowman et al. (2016), two parameter-tied sentence-encoding models are applied to the premise and the hypothesis sentences respectively, to generate the premise encoding $s^p$ and the hypothesis encoding $s^h$. A relation

---

[2]https://nlp.stanford.edu/projects/snli/

representation concatenating $s^p$, $s^h$, $s^p - s^h$ and $s^p \odot s^h$ is passed into a 300D fully connected layer, whose output is given to a 3-unit output layer with $\mathrm{softmax}$ to calculate the probability distribution over the three classes.

**Training Setup**: The optimization objective is the cross-entropy loss plus L2 regularization penalty. We minimize the objective by Adadelta (Zeiler, 2012) optimizer which is empirically more stable than Adam (Kingma & Ba, 2015) on SNLI. The batch size is set to 64 for all methods. The training phase takes 50 epochs to converge. All weight matrices are initialized by Glorot Initialization (Glorot & Bengio, 2010), and the biases are initialized with 0. We use 300D GloVe 6B pre-trained vectors (Pennington et al., 2014) to initialize the word embeddings in $x$. The Out-of-Vocabulary words in the training set are randomly initialized by uniform distribution between $(-0.05, 0.05)$. The word embeddings are fine-tuned during the training. The Dropout (Srivastava et al., 2014) keep probability and the L2 regularization weight decay factor $\gamma$ are set to $0.75$ and $5 \times 10^{-5}$, respectively. The number of hidden units is 300. The unspecified activation functions in all models are set to $\mathrm{Relu}$ (Glorot et al., 2011).

In Table 1, we report the number of parameters, and training/test accuracies of all baselines plus the methods from the official leaderboard. For fair comparison, we use 480D Bi-BloSAN, which leads to the similar parameter number with that of baseline encoders. Bi-BloSAN achieves the best test accuracy (similar to DiSAN) among all the sentence encoding models on SNLI. In particular, compared to the RNN models, Bi-BloSAN outperforms Bi-LSTM encoder, Bi-LSTM with attention and deep gated attention by 2.4%, 1.5% and 0.2%, respectively. Bi-BloSAN can even perform better than the semantic tree based models: SPINN-PI encoder (+2.5%)&Tree-based CNN encoder (+3.6%), and the memory network based model: NSE encoder (+1.1%). Additionally, Bi-BloSAN achieves the best performance among the baselines which are based on RNN/CNN/SAN. It outperforms Bi-LSTM (+0.7%), Bi-GRU (+0.8%), Bi-SRU (+0.9%), multi-CNN (+2.5%), Hrchy-CNN (+1.8%) and multi-head attention (+1.5%).

Table 2: Time cost and memory consumption of the different methods on SNLI. **Time(s)/epoch**: average training time (second) per epoch. **Memory(MB)**: Training GPU memory consumption (Megabyte). **Inference Time(s)**: average inference time (second) for all dev data on SNLI with test batch size of 100.

| Model | Time(s)/epoch | Memory(MB) | Inference Time(s) | Test Accuracy |
|---|---|---|---|---|
| Bi-LSTM (Graves et al., 2013) | 2080 | 1245 | 9.2 | 85.0 |
| Bi-GRU (Chung et al., 2014) | 1728 | 1259 | 9.3 | 84.9 |
| Bi-SRU (Lei & Zhang, 2017) | 1630 | 731 | 8.2 | 84.8 |
| Multi-CNN (Kim, 2014) | 284 | 529 | 2.4 | 83.2 |
| Hrchy-CNN (Gehring et al., 2017) | 343 | 2341 | 2.9 | 83.9 |
| Multi-head (Vaswani et al., 2017) | 345 | 1245 | 3.0 | 84.2 |
| DiSAN (Shen et al., 2017) | 587 | 2267 | 7.0 | 85.6 |
| 480D Bi-BloSAN | 508 | 1243 | 3.4 | 85.7 |

In addition, we compare time cost and memory consumption of all the baselines in Table 2. Compared to DiSAN with the same test accuracy, Bi-BloSAN is much faster and more memory efficient. In terms of training and inference time, Bi-BloSAN is $3 \sim 4\times$ faster than the RNN models (Bi-LSTM, Bi-GRU, etc.). It is as fast as CNNs and multi-head attention but substantially outperforms them in test accuracy. In terms of training memory, Bi-BloSAN requires similar GPU memory to the RNN-based models and multi-head attention, which is much less than that needed by DiSAN.

Finally, we conduct an ablation study of Bi-BloSAN in Table 3. In particular, we evaluate the contribution of each part of Bi-BloSAN by the change of test accuracy after removing the part. The removed part could be: 1) local context representations $h$, 2) global context representations $E$, 3) the context fusion module (mBloSA) or 4) all fundamental modules appeared in this paper. The results show that both the local and global context representations play significant roles in Bi-BloSAN. They make Bi-BloSAN surpass the state-of-the-art models. Moreover, mBloSA improves the test accuracy from 83.1% to 85.7%. Source2token self-attention performs much better than vanilla attention, and improves the test accuracy by 3.3%.

Table 3: An ablation study of Bi-BloSAN. "Local" denotes the local context representations $h$ and "Global" denotes the global context representations $E$. "Bi-BloSAN w/o mBloSA" equals to word embeddings directly followed by a source2token attention and "Bi-BloSAN w/o mBloSA & source2token self-attn." equals to word embeddings plus a vanilla attention without $q$.

| Model | $|\theta|$ | Test Accuracy |
|---|---|---|
| Bi-BloSAN | 2.8m | 85.7 |
| Bi-BloSAN w/o Local | 2.5m | 85.2 |
| Bi-BloSAN w/o Global | 1.8m | 85.3 |
| Bi-BloSAN w/o mBloSA | 0.54m | 83.1 |
| Bi-BloSAN w/o mBloSA & source2token self-attn. | 0.45m | 79.8 |

## 4.2 READING COMPREHENSION

Given a passage and a corresponding question, the goal of reading comprehension is to find the correct answer from the passage for the question. We use the Stanford Question Answering Dataset (Rajpurkar et al., 2016) (SQuAD)[3] to evaluate all models. SQuAD consists of questions posed by crowdworkers on a set of Wikipedia articles, where the answer to each question is a segment of text, or a span, from the corresponding passage.

Since Bi-BloSAN and other baselines are designed for sequence encoding, such as sentence embedding, we change the task from predicting the answer span to locating the sentence containing the correct answer. We build a network structure to test the power of sequence encoding in different models to find the correct answers. The details are given in Appendix B.

**Training Setup**: We use Adadelta optimizer to minimize the cross-entropy loss plus L2 regularization penalty, with batch size of 32. The network parameters and word embeddings initialization methods are same as those for SNLI, except that both the word embedding dimension and the number of hidden units are set to 100. We use 0.8 dropout keep probability and $10^{-4}$ L2 regularization weight decay factor.

We evaluate the Bi-BloSAN and the baselines except DiSAN because the memory required by DiSAN largely exceeds the GPU memory of GTX 1080Ti (11GB). The number of parameters, per epoch training time and the prediction accuracy on development set are given in Table 4.

Table 4: Experimental results for different methods on modified SQuAD task.

| Context Fusion Method | $|\theta|$ | Time(s)/Epoch | Dev Accuracy |
|---|---|---|---|
| Bi-LSTM (Graves et al., 2013) | 0.71m | 857 | 68.01 |
| Bi-GRU (Chung et al., 2014) | 0.57m | 782 | 67.98 |
| Bi-SRU (Lei & Zhang, 2017) | 0.32m | 737 | 67.32 |
| Multi-CNN (Kim, 2014) | 0.60m | 114 | 63.58 |
| Multi-head (Vaswani et al., 2017) | 0.45m | 140 | 64.82 |
| Bi-BloSAN | 0.82m | 293 | **68.38** |

Compared to RNN/CNN models, Bi-BloSAN achieves state-of-the-art prediction accuracy in this modified task. Bi-BloSAN shows its competitive context fusion and sequence encoding capability compared to Bi-LSTM, Bi-GRU, Bi-SRU but is much more time-efficient. In addition, Bi-BloSAN significantly outperforms multi-CNN and multi-head attention.

## 4.3 SEMANTIC RELATEDNESS

The goal of semantic relatedness is to predict the similarity degree of a given pair of sentences. Unlike the classification problems introduced above, predicting the semantic relatedness of sentences is a regression problem. We use $s^1$ and $s^2$ to denote the encodings of the two sentences, and assume that the similarity degree is between $[1, K]$. Following the method introduced by Tai et al. (2015), the concatenation of $s^1 \odot s^2$ and $|s^1 - s^2|$ is used as the representation of sentence related-

---

[3]https://rajpurkar.github.io/SQuAD-explorer/

ness. This representation is fed into a 300D fully connected layer, followed by a K-unit output layer with $\mathrm{softmax}$ to calculate a probability distribution $\hat{p}$. The details of this regression problem can be found in Appendix C. We evaluate all models on Sentences Involving Compositional Knowledge (SICK)[4] dataset, where the similarity degree is denoted by a real number in the range of $[1, 5]$. SICK comprises 9,927 sentence pairs with 4,500/500/4,927 instances for training/dev/test sets.

**Training Setup**: The optimization objective of this regression problem is the KL-divergence plus the L2 regularization penalty. We minimize the objective using Adadelta with batch size of $64$. The network parameters and word embeddings are initialized as in SNLI experiment. The keep probability of dropout is set to $0.7$, and the L2 regularization weight decay factor is set to $10^{-4}$.

Table 5: Experimental results for different methods on SICK sentence relatedness dataset. The reported accuracies are the mean of five runs (standard deviations in parentheses).

| Model | Pearson's $r$ | Spearman's $\rho$ | MSE |
|---|---|---|---|
| Meaning Factory (Bjerva et al., 2014) | 0.8268 | 0.7721 | 0.3224 |
| ECNU (Zhao et al., 2014) | 0.8414 | / | / |
| DT-RNN (Socher et al., 2014) | 0.7923 (0.0070) | 0.7319 (0.0071) | 0.3822 (0.0137) |
| SDT-RNN (Socher et al., 2014) | 0.7900 (0.0042) | 0.7304 (0.0042) | 0.3848 (0.0042) |
| Constituency Tree-LSTM (Tai et al., 2015) | 0.8582 (0.0038) | 0.7966 (0.0053) | 0.2734 (0.0108) |
| Dependency Tree-LSTM (Tai et al., 2015) | 0.8676 (0.0030) | 0.8083 (0.0042) | **0.2532 (0.0052)** |
| Bi-LSTM (Graves et al., 2013) | 0.8473 (0.0013) | 0.7913 (0.0019) | 0.3276 (0.0087) |
| Multi-CNN (Kim, 2014) | 0.8374 (0.0021) | 0.7793 (0.0028) | 0.3395 (0.0086) |
| Hrchy-CNN (Gehring et al., 2017) | 0.8436 (0.0014) | 0.7874 (0.0022) | 0.3162 (0.0058) |
| Multi-head (Vaswani et al., 2017) | 0.8521 (0.0013) | 0.7942 (0.0050) | 0.3258 (0.0149) |
| DiSAN (Shen et al., 2017) | **0.8695 (0.0012)** | **0.8139 (0.0012)** | 0.2879 (0.0036) |
| Bi-BloSAN | 0.8616 (0.0012) | 0.8038 (0.0012) | 0.3008 (0.0091) |

The performances of all models are shown in Table 5, which shows that Bi-BloSAN achieves state-of-the-art prediction quality. Although Dependency Tree-LSTM and DiSAN obtain the best performance, the Tree-LSTM needs external semantic parsing tree as the recursive input and expensive recursion computation, and DiSAN requires much larger memory for self-attention calculation. By contrast, Bi-BloSAN, as a RNN/CNN-free model, shows appealing advantage in terms of memory and time efficiency. Note that, performance of Bi-BloSAN is still better than some common models, including Bi-LSTM, CNNs and multi-head attention.

## 4.4 SENTENCE CLASSIFICATIONS

The goal of sentence classification is to correctly predict the class label of a given sentence in various scenarios. We evaluate the models on six sentence classification benchmarks for various NLP tasks, such as sentiment analysis and question-type classification. They are listed as follows.

- **CR**[5]: Customer reviews (Hu & Liu, 2004) of various products (cameras etc.). This task is to predict whether the review is positive or negative.

- **MPQA**[6]: Opinion polarity detection subtask of the MPQA dataset (Wiebe et al., 2005).

- **SUBJ**[7]: Subjectivity dataset (Pang & Lee, 2004), which includes a set of sentences. The corresponding label indicates whether each sentence is subjective or objective.

- **TREC**[8]: TREC question-type classification dataset (Li & Roth, 2002) which coarsely classifies the question sentences into six types.

---

[4] http://clic.cimec.unitn.it/composes/sick.html
[5] https://www.cs.uic.edu/~liub/FBS/sentiment-analysis.html
[6] http://mpqa.cs.pitt.edu
[7] https://www.cs.cornell.edu/people/pabo/movie-review-data/
[8] http://cogcomp.org/Data/QA/QC/

- **SST-1**[9]: Stanford Sentiment Treebank (Socher et al., 2013), which is a dataset consisting of movie reviews with five fine-grained sentiment labels, i.e., very positive, positive, neutral, negative and very negative.
- **SST-2**: Stanford Sentiment Treebank (Socher et al., 2013) with binary sentiment labels. Compared to SST-1, SST-2 removes the neutral instances, and labels the rest with either negative or positive.

Note that only SST-1 and SST-2 have the standard training/dev/test split, and TREC has the training/dev split. We implement 10-fold cross validation on SUBJ, CR and MPQA because the original datasets do not provide any split. We do not use the Movie Reviews (Pang & Lee, 2005) dataset because the SST-1/2 are extensions of it.

**Training Setup**: We use the cross-entropy loss plus L2 regularization penalty as the optimization objective. We minimize it by Adam with training batch size of 32 (except DiSAN, which uses batch size of 16 due to the limit of GPU memory). The network parameters and word embeddings are initialized as in SNLI experiment. To avoid overfitting on small datasets, we decrease the dropout keep probability and the L2 regularization weight decay factor $\gamma$ to 0.6 and $10^{-4}$, respectively.

Table 6: Experimental results for different methods on various sentence classification benchmarks. The reported accuracies on CR, MPQA and SUBJ are the mean of 10-fold cross validation, the accuracies on TREC are the mean of dev accuracies of five runs, and the accuracies on SST-1 and SST-2 are the mean of test accuracies of five runs. All standard deviations are in parentheses.

| Model | CR | MPQA | SUBJ | TREC | SST-1 | SST-2 |
|---|---|---|---|---|---|---|
| cBoW (Mikolov et al., 2013a) | 79.9 | 86.4 | 91.3 | 87.3 | / | / |
| Skip-thought (Kiros et al., 2015) | 81.3 | 87.5 | 93.6 | 92.2 | / | / |
| DCNN (Kalchbrenner et al., 2014) | / | / | / | 93.0 | 86.8 | 48.5 |
| AdaSent (Zhao et al., 2015) | 83.6 (1.6) | **90.4 (0.7)** | 92.2 (1.2) | 91.1 (1.0) | / | / |
| SRU (Lei & Zhang, 2017) | **84.8 (1.3)** | 89.7 (1.1) | 93.4 (0.8) | 93.9 (0.6) | **89.1 (0.3)** | / |
| Wide CNNs (Lei & Zhang, 2017) | 82.2 (2.2) | 88.8 (1.2) | 92.9 (0.7) | 93.2 (0.5) | 85.3 (0.4) | / |
| Bi-LSTM (Graves et al., 2013) | 84.6 (1.6) | 90.2 (0.9) | **94.7 (0.7)** | 94.4 (0.3) | 87.7 (0.6) | 49.9 (0.8) |
| Multi-head (Vaswani et al., 2017) | 82.6 (1.9) | 89.8 (1.2) | 94.0 (0.8) | 93.4 (0.4) | 83.9 (0.4) | 48.2 (0.6) |
| DiSAN (Shen et al., 2017) | **84.8 (2.0)** | 90.1 (0.4) | 94.2 (0.6) | 94.2 (0.1) | 87.8 (0.3) | **51.0 (0.7)** |
| Bi-BloSAN | **84.8 (0.9)** | **90.4 (0.8)** | 94.5 (0.5) | **94.8 (0.2)** | 87.4 (0.2) | 50.6 (0.5) |

The prediction accuracies of different models on the six benchmark datasets are given in Table 6. Bi-BloSAN achieves the best prediction accuracies on CR, MPQA and TREC, and state-of-the-art performances on SUBJ, SST-1 and SST-2 datasets (slightly worse than the best performances). Although Bi-BloSAN performs a little bit worse than the RNN models on SUBJ and SST-1, it is much more time-efficient than them. Additionally, on the SST-2 dataset, Bi-BloSAN performs slightly worse than DiSAN in terms of prediction accuracy ($-0.4\%$) but obtains a significantly higher memory utility rate.

We visualize the progress of training models on CR dataset in Figure 5. The convergence speed of Bi-BloSAN is $\sim 6\times$ and $\sim 2\times$ faster than Bi-LSTM and DiSAN respectively. Although Bi-BloSAN is less time-efficient than CNN and multi-head attention, it has much better prediction quality.

## 4.5 ANALYSES OF TIME COST AND MEMORY CONSUMPTION

To compare the efficiency-memory trade-off for each model on sequences of different lengths, we generate random tensor data, and feed them into the different sequence encoding models. The models we evaluate include Bi-LSTM, Bi-GRU, Bi-SRU, CNN, multi-head attention, DiSAN and Bi-BloSAN. The shape of the random data is [*batch size*, *sequence length*, *features number*]. We fix the *batch size* to 64 and the *features number* to 300, then change the *sequence length* from 16 to 384 with a step size 16.

We first discuss the time cost vs. the sequence length. As shown in Figure 6(a), the inference time of Bi-BloSAN is similar to those of multi-head attention and multi-CNN, but Bi-BloSAN outperforms

---
[9]http://nlp.stanford.edu/sentiment/

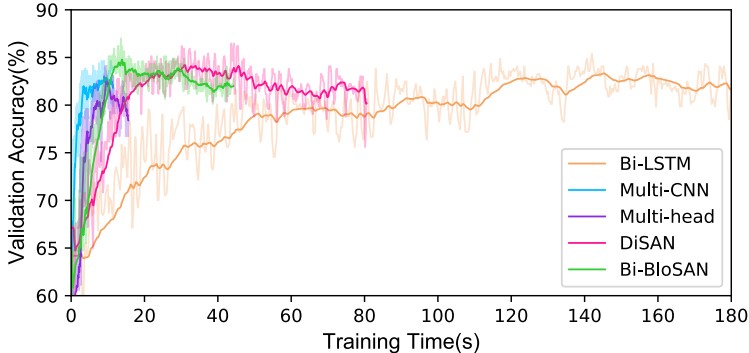

Figure 5: Validation accuracy vs. training time (second) of Bi-LSTM, CNN, multi-head attention, DiSAN and Bi-BloSAN for 800 training steps on CR dataset. (The Bi-LSTM for 800 steps consumes 279s in total.)

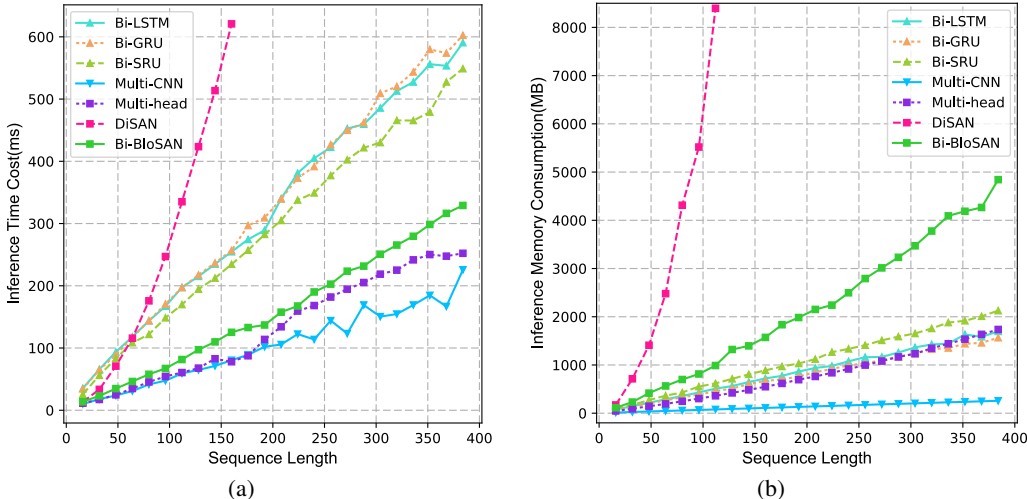

Figure 6: (a) Inference time cost and (b) GPU memory consumption of the sequence encoding models vs. the sequence length with the *batch size* of 64 and the *features number* of 300.

both by a large margin on prediction quality in previous experiments. Moreover, Bi-BloSAN is much faster than the RNN models (Bi-LSTM, Bi-GRU, BI-SRU). In addition, although DiSAN requires less training time than the RNN models in the experiments above, it is much slower during the inference phase because the large memory allocation consumes a great amount of time. By contrast, the block structure of Bi-BloSAN significantly reduces the inference time.

The GPU memory consumption vs. the sequence length for each model is visualized in Figure 6(b). DiSAN is not scalable because its memory grows explosively with the sequence length. Bi-BloSAN is more memory-efficient and scalable than DiSAN as the growth of its memory is nearly linear. Although Bi-BloSAN consumes more memory than the RNN models, it experimentally has better time efficiency and prediction quality. Since multi-head attention uses multiplicative attention, it requires less memory than all additive attention based models, such as DiSAN and Bi-BloSAN, but multiplicative attention based models usually perform worse than additive attention based models.

## 5    CONCLUSIONS

This paper presents an attention network, called bi-directional block self-attention network (Bi-BloSAN), for fast, memory-efficient and RNN/CNN-free sequence modeling. To overcome large memory consumption of existing self-attention networks, Bi-BloSAN splits the sequence into several blocks and employs intra-block and inter-block self-attentions to capture both local and long-

range context dependencies, respectively. To encode temporal order information, Bi-BloSAN applies forward and backward masks to the alignment scores between tokens for asymmetric self-attentions.

Our experiments on nine benchmark datasets for various different NLP tasks show that Bi-BloSAN can achieve the best or state-of-the-art performance with better efficiency-memory trade-off than existing RNN/CNN/SAN models. Bi-BloSAN is much more time-efficient than the RNN models (e.g., Bi-LSTM, Bi-GRU, etc.), requires much less memory than DiSAN, and significantly outperforms the CNN models and multi-head attention on prediction quality.

## 6 ACKNOWLEDGMENTS

This research was funded by the Australian Government through the Australian Research Council (ARC) under grants 1) LP160100630 partnership with Australia Government Department of Health and 2) LP150100671 partnership with Australia Research Alliance for Children and Youth (ARACY) and Global Business College Australia (GBCA). We also acknowledge the support of NVIDIA Corporation with the donation of GPU used for this research.

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

## A    THE SELECTION OF BLOCK LENGTH

In mBloSA, the length $r$ of each block is a hyper-parameter that determines memory consumption of mBloSA. To minimize the memory consumption, we propose an approach that calculates the optimized block length $r$ as follows.

We first introduce the method for determining $r$ for a dataset that has fixed sentence length $n$. Given the sentence length $n$ and the block number $m = n/r$, we have the following facts: 1) the major memory consumption in mBloSA is dominated by the masked self-attentions $g^m(\cdot, M)$; 2) the memory consumption of the masked self-attention is proportional to the square of the sentence length; and 3) mBloSA contains $m$ masked self-attention with a sequence length of $r$, and 1 masked

self-attention with a sequence length of $m$. Therefore, the memory $\xi$ required by mBloSA can be calculated by

$$\xi \propto r^2 \cdot m + m^2 \cdot 1$$
$$= r^2 \cdot \frac{n}{r} + (\frac{n}{r})^2. \tag{19}$$

By setting the gradient of $\xi$ w.r.t. $r$ to zero, we know that the memory consumption $\xi$ is minimum when $r = \sqrt[3]{2n}$.

Second, we propose a method for selecting $r$ given a dataset with the sentence lengths that follow a normal distribution $N(\mu, \sigma^2)$. We consider the case where mini-batch SGD with a batch size of $B$ or its variant is used for training. We need to calculate the upper bound of the expectation of the maximal sentence length for each mini-batch. Let us first consider $B$ random variables $[X_1, X_2, \ldots, X_B]$ in the distribution $N(0, \sigma^2)$. The goal is to find the upper bound of the expectation of random variable $Z + \mu$, where $Z$ is defined as

$$Z = \max_i X_i, \text{ for } i = 1, 2, \ldots, B. \tag{20}$$

By Jensen's inequality,

$$e^{t\mathbb{E}[Z]} \le \mathbb{E}[e^{tZ}] = E[\max_i e^{tX_i}]$$

$$\le \sum_{i=1}^{B} \mathbb{E}[e^{tX_i}] = ne^{t^2 \frac{\sigma^2}{2}} \tag{21}$$

Eq.(21) leads to

$$\mathbb{E}[Z] \le \frac{\ln B}{t} + \frac{t\sigma^2}{2}. \tag{22}$$

Let $t = \frac{\sqrt{2\ln B}}{\sigma}$ and we obtain the following upper bound.

$$\mathbb{E}[Z] \le \sigma\sqrt{2\ln B} \tag{23}$$

Hence, the upper bound of the expectation of the maximal sentence length among all the $B$ sentences in each mini-batch is $\sigma\sqrt{2\ln B} + \mu$. Therefore, the block length $r$ is computed by

$$r = \sqrt[3]{2n} = \sqrt[3]{2(\sigma\sqrt{2\ln B} + \mu)}.$$

## B  NETWORK SETUP FOR MACHINE COMPREHENSION

Each sample in the Stanford Question Answering Dataset (SQuAD) (Rajpurkar et al., 2016) is composed of three parts, i.e., a passage consisting of multiple sentences, a question sentence and a span in the passage indicating the position of the answer. In order to evaluate the performance of sentence embedding models, we change the task from predicting the span of the answer to finding the sentence containing the correct answer.

Given a passage consisting of $m$ sentences $[s^1, s^2, \ldots, s^m]$ where $s^k = [x_{k1}, x_{k2}, \ldots, x_{kn}]$, and the embedded question token sequence $q = [q_1, q_2, \ldots, q_l]$, the goal is to predict which sentence in the $m$ sentences contains the correct answer to the question $q$.

The neural net we use to evaluate different sequence encoding models is given in Figure 7. First, we process each sentence from the passage by a context fusion layer with shared parameters, followed by a source2token self-attention with shared parameters, which outputs a vector representation of the sentence. Therefore, the $m$ sentences are represented by $m$ vectors $[u_1, u_2, \ldots, u_m]$. The question sentence $q$ is compressed into a vector representation $q$ using source2token self-attention. Second, we combine each sentence $u_k$ with $q$ by concatenating $u_k$, $q$, $u_k - q$ and $u_k \odot q$, i.e.,

$$c_k = [u_k; q; u_k - q; u_k \odot q], \text{ for } k = 1, 2, \ldots, m. \tag{24}$$

Then $c = [c_1, c_2, \ldots, c_m]$ is fed into another context fusion layer that explores sentence-level dependencies. Finally, the resultant output representation of each sentence is separately fed into a fully connected layer to compute a scalar score indicating the possibility of the sentence containing the answer. A $\mathrm{softmax}$ function is applied to the scores of all $m$ sentences, to generate a probability distribution $\hat{p} \in \mathbb{R}^m$ for cross-entropy loss function. The sentence with the largest probability is predicted as the sentence containing the answer.

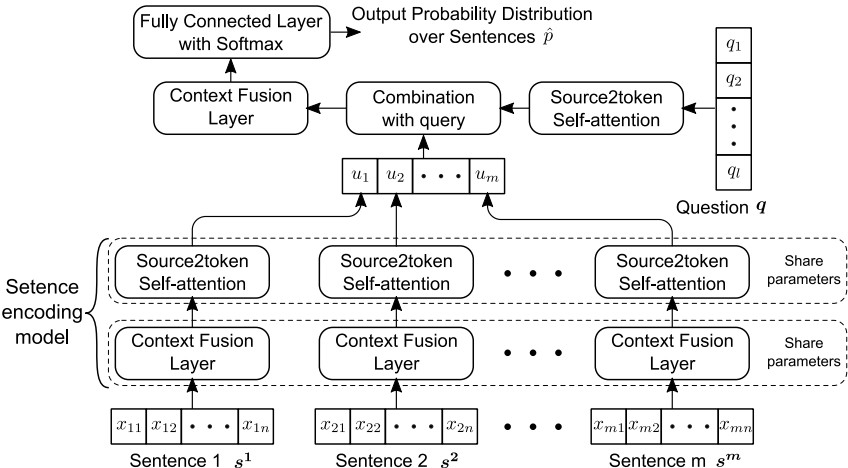

Figure 7: The structure of a neural network for machine comprehension. The candidates of the context fusion layer include Bi-LSTM, Bi-GRU, Bi-SRU, multi-CNN, multi-head attention and Bi-BloSA. Unlike the original multi-CNN for sentence embedding, we use padding and remove the max-pooling along the time axis to obtain an output of the same length as input. DiSA is not considered due to memory limitation.

## C  Loss of Regression Problem

Following the setting introduced by Tai et al. (2015) and given a predicted probability distribution $\hat{p}$ as the output of a feedforward network, the regression model predicts the similarity degree as

$$\hat{y} = \beta^T \hat{p}, \tag{25}$$

where $\beta = [1, 2, \ldots, K]$. The ground-truth similarity degree $y$ should be mapped to a probability distribution $p = [p_i]_{i=1}^{K}$ as the training target, where $p$ needs to fulfill $y = \beta^T p$. The mapping can be defined as

$$p_i = \begin{cases} y - \lfloor y \rfloor, & i = \lfloor y \rfloor + 1 \\ \lfloor y \rfloor - y + 1, & i = \lfloor y \rfloor \\ 0 & \text{otherwise} \end{cases}, \quad i = 1, 2, \ldots, K. \tag{26}$$

We use KL-divergence between $p$ and $\hat{p}$ as our loss function, i.e.,

$$L = \frac{1}{M} \sum_{k=1}^{M} KL(p^{(k)} || \hat{p}^{(k)}), \tag{27}$$

where the $p^{(k)}$ and $\hat{p}^{(k)}$ represent the target and predicted probability distributions of the $k$-th sample, respectively.

## D  Related Works

Recently, several structured attention mechanisms (Kim et al., 2017; Kokkinos & Potamianos, 2017) are proposed for capturing structural information from input sequence(s). When applied to self-attention, structured attentions share a similar idea to self-alignment attention (Hu et al., 2017) and multi-head attention (Vaswani et al., 2017) with one head, which aims to model the dependencies between the tokens. Similar to the attention from multiple perspectives in multi-head attention, multi-perspective context matching (Wang et al., 2016) explores the dependencies between passage and question from multiple perspectives for reading comprehension, while self-attentive structure (Lin et al., 2017) embeds sentences from various perspectives to produce matrix representations of the sentences. In recursive models, self-attention over children nodes (Teng & Zhang, 2017) can provide effective input for their parent node that has no standard input (Tai et al., 2015) as long as it is a non-leaf node in a semantic constituency parsing tree. Li et al. (2017) applies the multi-hop attention mechanism to transfer learning for cross-domain sentiment analysis without any RNN/CNN structure.

Bi-BloSA and hierarchical attention network (Yang et al., 2016) have similar structure, i.e., both stack two-layer attention mechanisms from bottom to top. However, they are different in three respects: 1) Bi-BloSA aims to learn context-aware representation for each token, while hierarchical attention is designed for document embedding; 2) the input of Bi-BloSA is a sentence, while the intput of hierarchical attention is a document composed of multiple sentences; and 3) the hierarchical attention performs vanilla attention twice, i.e., token-level attention on each sentence and sentence-level attention. Bi-BloSA, however, applies masked self-attention twice, i.e., intra-block self-attention and inter-block self-attention. Additionally, Bi-BloSA uses a feature fusion gate for each token to combine local and global context, and positional masks to encode temporal order information.

