# OpenReview forum: "Bi-Directional Block Self-Attention for Fast and Memory-Efficient Sequence Modeling"
_ICLR.cc/2018/Conference — Accept (Poster)_

### Official Review · AnonReviewer3 · 2017-11-27
**The methodology of the paper is incremental; the evaluation is comprehensive and in general supports the claims.**

**Rating:** 6
**Confidence:** 4

**Review:**

Pros:
The paper proposes a “bi-directional block self-attention network (Bi-BloSAN)” for sequence encoding, which inherits the advantages of multi-head (Vaswani et al., 2017) and DiSAN (Shen et al., 2017) network but is claimed to be more memory-efficient. The paper is written clearly and is easy to follow. The source code is released for duplicability. The main originality is using block (or hierarchical) structures; i.e., the proposed models split the an entire sequence into blocks, apply an intra-block SAN to each block for modeling local context, and then apply an inter-block SAN to the output for all blocks to capture long-range dependency. The proposed model was tested on nine benchmarks  and achieve good efficiency-memory trade-off.

Cons:
- Methodology of the paper is very incremental compared with previous models.
- Many of the baselines listed in the paper are not competitive; e.g.,  for SNLI, state-of-the-art results are not included in the paper.
- The paper argues advantages of the proposed models over CNN by assuming the latter only captures local dependency, which, however, is not supported by discussion on or comparison with hierarchical CNN.
- The block splitting (as detailed in appendix) is rather arbitrary in terms of that it potentially divides coherent language segments apart. This is unnatural, e.g., compared  with alternatives such as using linguistic segments as blocks.
- The main originality of paper is the block style. However, the paper doesn’t analyze how and why the block brings improvement.
-If we remove intra-block self-attention (but only keep token-level self-attention), whether the performance will be significantly worse?

---

> ### Author Response · Authors · 2017-12-12
> **Memory may not be reduced effectively if using linguistic segments; removing intra-block self-attention decreases the performance.**
>
> ==Following above==
> - Q4. The block splitting (as detailed in appendix) is rather arbitrary in terms of that it potentially divides coherent language segments apart. This is unnatural, e.g., compared with alternatives such as using linguistic segments as blocks.
>
> Here are two reasons for not using linguistic segments as blocks in our model. Firstly, the property of significantly reducing memory cannot be guaranteed if using linguistic segments, because either too long or too short segments will lead to expensive memory consumption, and we cannot easily control the length of linguistic segments provided by other tools. For example, in Eq.(19), either large and small block length r is likely to result in large memory. Secondly, the process of achieving linguistic segments potentially increases computation/memory cost, introduces overhead and requires more complex implementation. In addition, although we do not use linguistic segments for block splitting, our model can still capture the dependencies between tokens from different blocks by using the block-level context fusion and feature fusion gate developed in this paper.
>
>
> - Q5. The main originality of paper is the block style. However, the paper doesn’t analyze how and why the block brings improvement.
>
> The block or two-layer self-attention substantially reduces the memory and computational costs required by previous self-attention mechanisms, which is proportional to the square of sequence length. Meanwhile, it achieves competitive or better accuracy than RNNs/CNNs. We give a formally explanation of how this block idea can reduce the memory in Appendix A.
>
>
> - Q6. If we remove intra-block self-attention (but only keep token-level self-attention), whether the performance will be significantly worse?
>
> Compared to test accuracy 85.7% of Bi-BloSAN on SNLI, the accuracy will be decreased to 85.2% if we remove the intra-block attention (keep block-level attention), whereas the accuracy will be decreased to 85.3% if we remove inter-block self-attention (keep token-level self-attention in blocks). Moreover, if we only use token-level self-attention, the model will be identical to the directional self-attention [2]. You can refer to the ablation study at the end of Section 4.1 for more details.
>
>
>
> References
> [1] Vaswani, Ashish, et al. "Attention is all you need. CoRR abs/1706.03762." (2017).
> [2] Shen, Tao, et al. "Disan: Directional self-attention network for rnn/cnn-free language understanding." arXiv preprint arXiv:1709.04696 (2017).
> [3] Srivastava, Rupesh Kumar, Klaus Greff, and Jürgen Schmidhuber. "Highway networks." arXiv preprint arXiv:1505.00387 (2015).
> [4] Nie, Yixin, and Mohit Bansal. "Shortcut-stacked sentence encoders for multi-domain inference." arXiv preprint arXiv:1708.02312 (2017).
> [5] Jihun Choi, Kang Min Yoo and Sang-goo Lee. "Learning to compose task-specific tree structures." arXiv preprint arXiv:1707.02786 (2017).
> [6]Kim, Yoon. "Convolutional neural networks for sentence classification." arXiv preprint arXiv:1408.5882 (2014).
> [7] Kaiser, Łukasz, and Samy Bengio. "Can Active Memory Replace Attention?." Advances in Neural Information Processing Systems. 2016.
> [8] Kalchbrenner, Nal, et al. "Neural machine translation in linear time." arXiv preprint arXiv:1610.10099 (2016).
> [9] Gehring, Jonas, et al. "Convolutional Sequence to Sequence Learning." arXiv preprint arXiv:1705.03122 (2017).

---

> ### Author Response · Authors · 2017-12-12
> **Novel context fusion is developed for the two-level self-attention; Bi-BloSAN achieves the best accuracy among all sentence-encoding models on SNLI; Hierarchical CNN is costly for long-range dependency.**
>
> Thank you for your elaborative comments! We discuss the Cons you pointed out one by one as follows.
>
> - Q1. Methodology of the paper is very incremental compared with previous models.
>
> Yes, the idea of using block or two-level attention is simple. In fact, it is similar to the idea behind almost all the hierarchical models. However, it has never been studied on self-attentions based models, especially on attention-only models (as much as we know, Transformer [1] and DiSAN [2] are the merely two published attention-only models), for context fusion. Moreover, it solves a critical problem of previous self-attention mechanisms, i.e., expensive memory consumption, which was a burden of applying attention to long sequences and an inevitable weakness compared to popular RNN models. Hence, it is a simple idea, which leads to a simple model, but effectively solves an important problem.
>
> In addition, given this idea, it is non-trivial to design a neural net architecture for context fusion, we still need to figure out: 1) How to split the sequence so the memory can be effectively reduced? 2) How to capture the dependency between two elements from different blocks? 3) How to produce contextual-aware representation for each element on each level? 4) How to combine the output of different levels so the information from lower level does not fade out? For example, on top of Figure 3, we duplicate the block features e_i to each element as its high-level representation, use skip (highway [3]) connections to achieve its lower level representations x_i and h_i, and then design a fusion gate to combine the three representations. This design assigns each element with both high-level and low-level representations and combine them on top of the model to produce a contextual-aware representation per input element. Without it, the two-level attention can only give us e_i, which cannot explicitly model the dependency between elements from different blocks, and cannot be used for context fusion. This method has not been used in construction of attention-based models because multi-level self-attention had not been studied before.
>
>
> - Q2. Many of the baselines listed in the paper are not competitive; e.g., for SNLI, state-of-the-art results are not included in the paper.
>
> In the experiment on SNLI, Bi-BloSAN is only used to produce sentence encoding. For a fair comparison, we only compare it with the sentence-encoding based models listed separately on the leaderboard of SNLI. Up to ICLR submission deadline, Bi-BloSAN achieves the best test accuracy among all of them.
>
> After ICLR submission deadline, the leaderboard has been updated with several new methods. We copy the results of the new methods in the following.
> The Proposed Model) 480D Bi-BloSAN	2.8M	85.7%
> 1) 300D Residual stacked encoders[4]	9.7M	85.7%
> 2) 600D Gumbel TreeLSTM encoders[5]	10.0M	86.0%
> 3) 600D Residual stacked encoders[4]	29.0M	86.0%
> These results show that compared to the newly updated methods, Bi-BloSAN uses significantly less parameters but achieves competitive test accuracy.
>
>
> - Q3. The paper argues advantages of the proposed models over CNN by assuming the latter only captures local dependency, which, however, is not supported by discussion on or comparison with hierarchical CNN.
>
> The discussion about CNN in the current version mainly focuses on single layer CNN with multi-window [6], which is widely used in NLP community, and does not mention too much about recent studies on hierarchical CNNs. The hierarchical CNNs in NLP, such as Extended Neural GPU [7], ByteNet [8], and ConvS2S [9], are able to model relatively long-range dependency by using stacking CNNs, which can increase the number of input elements represented in a state. Nonetheless, as mentioned in [1], the number of operations (i.e. CNNs) required to relate signals from two arbitrary input grows in the distance between positions, linearly for ConvS2S and logarithmically for ByteNet. This makes it more difficult to learn dependencies between distant positions. However, self-attention based method only requires constant number of operations, no matter how far it is between two elements. We will add the discussion on hierarchical CNNs in the revision.

---

> ### Author Response · Authors · 2017-12-18
> **More experiments show hierarchical CNN does not perform well on SNLI**
>
> To test the performance of hierarchical CNN for context fusion, we implemented it on SNLI dataset. In particular, we used 3-layer 300D CNNs with kernel length 5 (i.e., using n-gram of n=5). By following [1], we also applied "Gated Linear Units (GLU)" [2] and residual connection [3] to the hierarchical CNN. We tuned the keep probability of dropout between 0.65 and 0.85 with step-size 0.05. The code of this hierarchical CNNs can be found at https://github.com/code4review/BiBloSA/blob/master/context_fusion/hierarchical_cnn.py
>
> This model has 3.4M parameters. It spends 343s per training epoch and 2.9s for inference on dev set. Its test accuracy is 83.92% (with dev accuracy 84.15% and train accuracy 91.28%), which slightly outperforms the CNNs with multi-window [4] shown in our paper, but is still worse than other baselines and Bi-BloSAN. We will add these results to the revision.
>
> [1] Gehring, Jonas, et al. "Convolutional Sequence to Sequence Learning." arXiv preprint arXiv:1705.03122 (2017).
> [2] Dauphin, Yann N., et al. "Language modeling with gated convolutional networks." arXiv preprint arXiv:1612.08083 (2016).
> [3] He, Kaiming, et al. "Deep residual learning for image recognition." Proceedings of the IEEE conference on computer vision and pattern recognition. 2016.
> [4] Kim, Yoon. "Convolutional neural networks for sentence classification." arXiv preprint arXiv:1408.5882 (2014).

---

### Official Review · AnonReviewer4 · 2017-11-27
**Strong support for more efficient attention**

**Rating:** 9
**Confidence:** 4

**Review:**

This high-quality paper tackles the quadratic dependency of memory on sequence length in attention-based models, and presents strong empirical results across multiple evaluation tasks. The approach is basically to apply self-attention at two levels, such that each level only has a small, fixed number of items, thereby limiting the memory requirement while having negligible impact on speed. It captures local information into so-called blocks using self-attention, and then applies a second level of self-attention over the blocks themselves.

The paper is well organized and clearly written, modulo minor language mistakes that should be easy to fix with further proof-reading. The contextualization of the method relative to CNNs/RNNs/Transformers is good, and the beneficial trade-offs between memory, runtime and accuracy are thoroughly investigated, and they're compelling.

I am curious how the story would look if one tried to push beyond two levels...? For example, how effective might a further inter-sentence attention level be for obtaining representations for long documents?

Minor points:
- Text between Eq 4 & 5: W^{(1)} appears twice; one instance should probably be W^{(2)}.
- Multiple locations, e.g. S4.1: for NLI, the word is *premise*, not *promise*.
- Missing word in first sentence of S4.1: ... reason __ the ...

---

> ### Author Response · Authors · 2017-12-12
> **Thanks for your strong support! Extending our two-level self-attention to multi-level is worth studying for long documents.**
>
> Thank you for your strong support to our work! We will carefully fix the typos you pointed out.
>
> - Q1. I am curious how the story would look if one tried to push beyond two levels...? For example, how effective might a further inter-sentence attention level be for obtaining representations for long documents?
>
> We have different answers to this question for sequences with different lengths.
>
> For context fusion or embedding of single sentences (which is the main focus of this paper), a two-level self-attention is usually sufficient to reduce the memory consumption and meanwhile to inherit most power of original SAN in modeling contextual dependencies. Compared to multi-level attention, it preserves the local dependencies in longer subsequence and directly controls the memory utility rate, by using less parameters and computations than multi-level one.
>
> For the context fusion of a document or a passage, which already has a multi-level structure (document-passages-sentences-phrases), it is worth considering to use multi-level self-attention to model the contextual relationship when the memory consumption needs to be small. Recently, self-attention has been applied to long text as a popular context fusion strategy in machine comprehension task [1,2]. In this task, the original self-attention requires lots of memory, and cannot be solely applied due to the difficulty of context fusion for a long passage/document. It is more practical to use LSTM or GRU as context fusion layers and use self-attention as a complementary module capturing the distance-irrelevant dependency. But the recurrent structure of LSTM/GRU leads to inefficiency in computation. Therefore, multi-level self-attention could provide a both memory and time efficient solution. For example, we can design a three-level self-attention structure, which consists of intra-block intra sentence, inter-block intra sentence and inter-sentence self-attentions, to produce context-aware representations of tokens from a passage. Such model can overcome the weaknesses of both RNN/CNN-based SANs (only used as a complimentary module to context fusion layers) and the RNN/CNN-free SANs (with explosion of memory consumption when text length grows).
>
>
>
> References
> [1] Hu, Minghao, Yuxing Peng, and Xipeng Qiu. "Reinforced mnemonic reader for machine comprehension." CoRR, abs/1705.02798 (2017).
> [2] Huang, Hsin-Yuan, et al. "FusionNet: Fusing via Fully-Aware Attention with Application to Machine Comprehension." arXiv preprint arXiv:1711.07341 (2017).

---

### Official Review · AnonReviewer2 · 2017-11-30
**solid experiments, but the model is not very exciting**

**Rating:** 6
**Confidence:** 4

**Review:**

This paper introduces bi-directional block self-attention model (Bi-BioSAN) as a general-purpose encoder for sequence modeling tasks in NLP. The experiments include tasks like natural language inference, reading comprehension (SquAD), semantic relatedness and sentence classifications. The new model shows decent performance when comparing with Bi-LSTM, CNN and other baselines while running at a reasonably fast speed.

The advantage of this model is that we can use little memory (as in RNNs) and enjoy the parallelizable computation as in (SANs), and achieve similar (or better) performance.

While I do appreciate the solid experiment section, I don't think the model itself is sufficient contribution for a publication at ICLR. First, there is not much innovation in the model architecture. The idea of the Bi-BioSAN model simply to split the sentence into blocks and compute self-attention for each of them, and then using the same mechanisms as a pooling operation followed by a fusion level. I think this more counts as careful engineering of the SAN model rather than a main innovation. Second, the model introduces much more parameters. In the experiments, it can easily use 2 times parameters than the commonly used encoders. What if we use the same amount of parameters for Bi-LSTM encoders? Will the gap between the new model and the commonly used ones be smaller?

====

I appreciate the answers the authors added and I change the score to 6.

---

> ### Author Response · Authors · 2017-12-12
> **Novel context fusion is developed for the two-level self-attention; Bi-BloSAN still outperforms Bi-LSTM when having the similar number of parameters.**
>
> Thanks for your comments!
>
> - Q1. First, there is not much innovation in the model architecture. The idea of the Bi-BioSAN model simply to split the sentence into blocks and compute self-attention for each of them, and then using the same mechanisms as a pooling operation followed by a fusion level. I think this more counts as careful engineering of the SAN model rather than a main innovation.
>
> Yes, the idea of using block or two-level attention is simple. In fact, it is similar to the idea behind almost all the hierarchical models. However, it has never been studied on self-attentions based models, especially on attention-only models (as much as we know, Transformer [1] and DiSAN [2] are the merely two published attention-only models), for context fusion. Moreover, it solves a critical problem of previous self-attention mechanisms, i.e., expensive memory consumption, which was a burden of applying attention to long sequences and an inevitable weakness compared to popular RNN models. Hence, it is a simple idea, which leads to a simple model, but effectively solves an important problem.
>
> In addition, given this idea, it is non-trivial to design a neural net architecture for context fusion, we still need to figure out: 1) How to split the sequence so the memory can be effectively reduced? 2) How to capture the dependency between two elements from different blocks? 3) How to produce contextual-aware representation for each element on each level? 4) How to combine the output of different levels so the information from lower level does not fade out? For example, on top of Figure 3, we duplicate the block features e_i to each element as its high-level representation, use skip (highway [3]) connections to achieve its lower level representations x_i and h_i, and then design a fusion gate to combine the three representations. This design assigns each element with both high-level and low-level representations and combine them on top of the model to produce a contextual-aware representation per input element. Without it, the two-level attention can only give us e_i, which cannot explicitly model the dependency between elements from different blocks, and cannot be used for context fusion. This method has not been used in construction of attention-based models because multi-level self-attention had not been studied before.
>
>
> - Q2. Second, the model introduces much more parameters. In the experiments, it can easily use 2 times parameters than the commonly used encoders. What if we use the same amount of parameters for Bi-LSTM encoders? Will the gap between the new model and the commonly used ones be smaller?
>
> As suggested by you, we studied two cases in which Bi-LSTM and Bi-BloSAN have similar number of parameters. The gap does not change in both cases. We will add these new results to our revision.
>
> 1) We increase the number of hidden units in Bi-LSTM encoders from 600 to 800. This increases the number of parameters from 2.9M to 4.8M, which is more than 4.1M of Bi-BloSAN. We implement this 800D Bi-LSTM encoder on the SNLI dataset which is the largest benchmark dataset used in this paper. After tuning of the hyperparameters (e.g., dropout keep probability is increased from 0.65 to 0.80 with step 0.05 in case of overfitting), the best test accuracy is 84.95% (with dev accuracy of 85.67%).
>
> 2) We decrease the number of hidden units in Bi-BloSAN from 600 to 480. This reduces the number of parameters from 4.1M to 2.8M, which is similar to that of the commonly used encoders. Interestingly, without tuning the keep probability of dropout, the test accuracy of this 480D Bi-BloSAN is 85.66% (with dev accuracy 86.08% and train accuracy 91.68%).
>
> Additionally, a recent NLP paper [4] shows that increasing the dimension of an RNN encoder from 128D to 2048D does not result in substantially improvement of the performance (from 21.50 to 21.86 of BLEU score on newstest2013 for machine translation). This is consistent with the results above.
>
>
>
> References
> [1] Vaswani, Ashish, et al. "Attention is all you need. CoRR abs/1706.03762." (2017).
> [2] Shen, Tao, et al. "Disan: Directional self-attention network for rnn/cnn-free language understanding." arXiv preprint arXiv:1709.04696 (2017).
> [3] Srivastava, Rupesh Kumar, Klaus Greff, and Jürgen Schmidhuber. "Highway networks." arXiv preprint arXiv:1505.00387 (2015).
> [4] Britz, Denny, et al. "Massive exploration of neural machine translation architectures." arXiv preprint arXiv:1703.03906 (2017).

---

### Author Response · Authors · 2017-12-12
**Summary of Revision-V1**

Dear all reviewers, we upload a revision of this paper that differs from the previous one in that
1) We found the multi-head attention is very sensitive to the keep probability of dropout due to "Attention Dropout", so we tuned it in interval [0.70:0.05:0.90], resulting in test accuracy on SNLI increasing from 83.3% to 84.2%.
2) As suggested by AnonReviewer2, we decreased the hidden units number of Bi-BliSAN from 600 to 480 on SNLI, which leads to the parameters number dropping from 4.1M to 2.8M. The test accuracy of this 480D Bi-BloSAN is 85.66% with dev accuracy 86.08% and train accuracy 91.68%.
3) As suggested by AnonReviewer3, we added the discussion on hierarchical CNNs to the introduction.
4) We corrected typos and mistakes partly pointed out by AnonReviewer4.

---

### Author Response · Authors · 2017-12-27
**Summary of Revision-V2**

Dear all reviewers, we upload a revision of this paper that differs from the previous one in that
1) As suggested by AnonReviewer3, we implemented the Hierarchical CNN (called Hrchy-CNN in the paper) as a baseline, and we then applied this model to SNLI and SICK datasets, which showed that the proposed model, Bi-BloSAN, still outperforms the Hierarchical CNN by a large margin;
2) We fixed some typos.

---

### Public Comment · ~shen_si_zhe1 · 2018-12-02
**I couldn't get the same result as the paper...**

This paper introduces bi-directional block self-attention model (Bi-BioSAN) as a general-purpose encoder for sequence modeling tasks in NLP.

For example ,when I use the cr dataset,

"python sc_main.py --network_type exp_context_fusion --context_fusion_method wblock --model_dir_suffix training --dataset_type cr --gpu 0 "

the result is not the 84.48 as the paper,I could only get 84.30 after several times.
 I need your help!
Thank you !

---

> ### Author Response · Authors · 2018-12-02
> **Reply**
>
> 1. Try to tune the dropout probability of the neural network because CR is a small-scale dataset, and you can also tune the block length for better performance when implemented on a small dataset.
> 2. As the similar issue occurring in my github repo https://github.com/taoshen58/BiBloSA/issues/2 you can refer to that for the solutions.

---

### Public Comment · (anonymous) · 2018-12-14
**一些问题**

您好，我想请教一下。您的文章中经常提到GPU Memory 这一项指标，但是我很疑惑，如何计算或是通过编写代码获得这一数值。请问，您是如何做到的？
谢谢您。

---

### Decision · Program_Chairs · 2018-01-29
**ICLR 2018 Conference Acceptance Decision**

**Decision:**

Accept (Poster)

**Comment:**

The proposed Bi-BloSAN is a two-levels' block SAN, which has both parallelization efficiency and memory efficiency. The study is thoroughly conducted and well presented.